# Insight into the Postbiotic Potential of the Autochthonous Bacteriocin-Producing *Enterococcus faecium* BGZLM1-5 in the Reduction in the Abundance of *Listeria monocytogenes* ATCC19111 in a Milk Model

**DOI:** 10.3390/microorganisms11122844

**Published:** 2023-11-23

**Authors:** Nikola Popović, Dušan Stevanović, Dušan Radojević, Katarina Veljović, Jelena Đokić, Nataša Golić, Amarela Terzić-Vidojević

**Affiliations:** Institute of Molecular Genetics and Genetic Engineering, University of Belgrade, Vojvode Stepe 444a, 11042 Belgrade, Serbia; dstevanovic@imgge.bg.ac.rs (D.S.); dradojevic@imgge.bg.ac.rs (D.R.); katarinav@imgge.bg.ac.rs (K.V.); jelena.djokic@imgge.bg.ac.rs (J.Đ.); natasag@imgge.bg.ac.rs (N.G.); amarela@imgge.bg.ac.rs (A.T.-V.)

**Keywords:** enterococci, probiogenomics, bacteriocin, safety, *Listeria* spp., milk model

## Abstract

This study aimed to explore the probiogenomic characteristics of artisanal bacteriocin-producing *Enterococcus faecium* BGZLM1-5 and its potential application in reducing *Listeria monocytogenes* in a milk model. The BGZLM1-5 strain was isolated from raw cow’s milk from households in the Zlatar Mountain region. The whole genome sequencing approach and bioinformatics analyses reveal that the strain BGZLM1-5 is non-pathogenic to humans. Bacteriocin-containing supernatant was thermally stable and antimicrobial activity retained 75% of the initial activity compared with that of the control after treatment at 90 °C for 30 min. Antimicrobial activity maintained relative stability at pH 3–11 and retained 62.5% of the initial activity compared with that of the control after treatment at pH 1, 2, and 12. The highest activity of the partially purified bacteriocin was obtained after precipitation at 40% saturation with ammonium sulfate and further purification by mixing with chloroform. Applying 3% and 5% (*v/v*) of the bacteriocin-containing supernatant and 0.5% (*v*/*v*) of the partially purified bacteriocin decreased the viable number of *L. monocytogenes* ATCC19111 after three days of milk storage by 23.5%, 63.5%, and 58.9%, respectively.

## 1. Introduction

*Listeria monocytogenes* is an important foodborne pathogen frequently isolated from a variety of food products and causing listeriosis, one of the most serious and severe food-borne diseases in humans [1,2]. The reduction of the viable count of *Listeria* spp. in food and the environment can be achieved by the antimicrobial activity of lactic acid bacteria (LAB), especially by the action of enterocins produced by certain strains of enterococci [3]. Numerous *Enterococcus faecium* strains inhibit the growth of pathogens, such as *Listeria* spp., by producing antimicrobial compounds, and they were recognized for their probiotic benefits many years ago [4,5]. Furthermore, enterococci have the potential to be used as natural preservatives in food products owing to the possibility for producing various proteinaceous and non-proteinaceous antimicrobial compounds [6,7]. In addition to probiotics, postbiotics are recognized as the preparation of inanimate microorganisms and/or their components that confer a health benefit on the host [8]. Thus, enterococcal products can be considered as a safe way to exploit the natural characteristics of different enterococcal strains.

Most enterococcal bacteriocins of group II are thermostable and resistant to various technological challenges, making them suitable for use in the food industry [9]. They remain active at low temperatures for prolonged periods and are heat resistant. These properties enable them to effectively preserve fermented, refrigerated, and pasteurized foods. Additionally, all Class II bacteriocins possess noteworthy traits relevant to food safety technology, including resistance to extreme pH levels, temperature variations, and salinity [10]. Although the use of bacteriocins is considered safe, in most cases, the use of bacteriocins is not successful because of the degradation or adsorption of bacteriocins into complex food matrices [11]. Only nisin produced by *Lactococcus lactis* subsp. *lactis* is generally accepted for use in the food industry; however, owing to nisin’s low activity in non-acidic environments, its application is limited [12].

Autochthonous *En. faecium* BGZLM1-5 originated from raw cow’s milk collected from a household on Zlatar Mountain, Republic of Serbia [13]. Because we previously demonstrated the strong antilisterial effect of live BGZLM1-5 [14], this study aimed to investigate the possibility of reducing the abundance of *L. monocytogenes* ATCC19111 with a postbiotic produced by the strain BGZLM1-5 as a safer way to use the natural potential of this strain. Additionally, bioinformatical analyses of the sequenced genome provided insight into the probiotic properties of the strain.

## 2. Materials and Methods

### 2.1. Bacterial Strains and Growth Conditions

The bacterial strains used in this study are listed in Table 1. *Enterococcus faecium* BGZLM1-5, *En. faecium* DDE4, *L. monocytogenes* ATCC19111, *Listeria ivanovii* ATCC19119, and *Listeria innocua* ATCC33090 were grown in M17 broth (Merck GmbH, Darmstadt, Germany) supplemented with glucose (0.5% *w*/*v*) (GM17 broth) at 37 °C for 24 h under aerobic conditions. The other bacteria mentioned in Table 1 were cultivated in Luria-Bertani broth (LB) (Torlak, Belgrade, Serbia), containing 0.5% NaCl, 0.5% yeast extract (Torlak), and 1% tryptone (Torlak) at 37 °C under aerobic conditions. Corresponding agar plates were prepared by adding agar (1.7% *w*/*v*) (Torlak) to each broth.

### 2.2. Whole Genome Sequencing, Standard Analytical Procedures, and Data Submission

The genomic DNA of *En. faecium* BGZLM1-5 was extracted using the previously described method [19]. The quality and concentration of the total DNA were assessed through agarose gel electrophoresis and a Bio-Spec nano spectrophotometer (Shimadzu, Tokyo, Japan). The whole genome sequencing (WGS) of the BGZLM1-5 genome was performed at MicrobesNG service (MicrobesNG, IMI-School of Biosciences, University of Birmingham, Birmingham, UK). The Nextera XT Library Prep Kit protocol (Illumina, San Diego, CA, USA) was used for the library preparation, with a slight modification to the manufacturer’s instructions, using 2 ng of input DNA, and the PCR elongation time was increased to 1 min. The library preparation was performed on a Hamilton Microlab STAR (Hamilton, Reno, NV, USA). Pooled libraries were quantified on a Roche light cycler 96 qPCR machine (Roche, Indianapolis, IN, USA) using the Kapa Biosystems Library Quantification Kit for Illumina. Sequencing was performed on the Illumina HiSeq 2500 platform utilizing a 2 × 250 bp paired-end approach. Adapter trimming was executed using Trimmomatic v0.30 [20] with a sliding window quality cutoff set at Q15. De novo assembly was carried out using SPAdes v3.7 [21], followed by contig annotation utilizing Prokka 1.11 [22]. The Kraken 1 software [23] was employed to determine the closest reference genome available. BWA mem [24] was used to align the reads to the reference genome. Genome circular maps were drawn using the Proksee online tool with default parameters [25]. The preliminary genome sequences of BGZLM1-5 have been submitted to GenBank under the accession JAVDBU000000000. The version described in this paper is version JAVDBU010000000.

### 2.3. Genotypic Characterization for Safety and Probiotic-Related Traits

The obtained contigs were employed to ascertain the resistome, using the ResFinder 4.1, Pathogen-Finder v1.1, and PlasmidFinder v2.1.1 tools, accessible at the Center for Genomic Epidemiology (www.genomicepidemiology.org; accessed on 26 April 2023.), with default configurations. Additionally, the Virulence Factor of Bacterial Pathogens Database (VFDB) (http://www.mgc.ac.cn/VFs/main.htm; accessed on 26 April 2023.) [26] was utilized to predict potential virulence genes. In terms of beneficial traits, the complete genome of BGZLM1-5 underwent analysis with BAGEL4 (http://bagel4.molgenrug.nl/; accessed on 26 April 2023.) [27] to identify possible antimicrobial compounds.

### 2.4. Gene Expression Analysis by Reverse-Transcription Quantitative PCR

The overnight cultures were diluted in fresh GM17 medium (1% *v*/*v*) and incubated at 37 °C. The total RNA from BGZLM1-5 cells was isolated using the RNeasy Mini Kit (Qiagen, Hilden, Germany), with a modified lysis step at 0 h, 6 h, and 24 h. Reverse transcription was conducted with a RevertAid RT Reverse Transcription Kit (Thermo Fisher Scientific, Austin, TX, USA) according to the manufacturer’s protocol. The primers used in qPCR are listed in Table 2. qPCR was performed with the FastGene IC Green 2× PCR Universal Mix (Nippon Genetics Europe GmbH, Düren, Germany) in a 7500 real-time PCR machine (Applied Biosystems, Lincoln Centre Drive, Foster City, CA, USA) under the following conditions: 2 min at 95 °C activation, 40 cycles of 5 s at 95 °C, and 30 s at 60 °C. Normalization was conducted using the ^∆∆^CT method [28]. The qPCR experiments were conducted in triplicate. All the results are represented as mean values ± standard deviations obtained from three independent experiments.

### 2.5. Antimicrobial Activity of Enterococcus faecium BGZLM1-5

The antimicrobial activity of *En. faecium* BGZLM1-5 against the bacteria listed in Table 1 was tested using the agar well diffusion method [3]. Briefly, a 50 μL aliquot of the sample (overnight culture) was examined in pre-made soft agar wells (7 mm in diameter), and indicator plates were incubated under appropriate conditions for the respective indicator strain for 24 h. A crystal of pronase E (Sigma, St. Louis, MO, USA) was used as a control for the proteinaceous nature of antimicrobial compounds.

### 2.6. Preparation of Enterococcus faecium BGZLM1-5 Postbiotic

The preparation of the postbiotic was conducted according to the previously described protocol [3]. The supernatant (SN BGZLM1-5) of the overnight (ON) BGZLM1-5 culture was centrifuged for 10 min at 5000 rpm, collected, filtrated through 0.22 μm membrane filters (Sartorius, Goettingen, Germany), and stored at −20 °C. Live BGZLM1-5 cells were washed two times with phosphate-buffered saline (PBS, pH 7), concentrated ten times in PBS, and used for further assays (ON BGZLM1-5). Parts of the live-cell suspension were heated at 60, 70, 80, 90, and 100 °C for 30 min to obtain heat-killed BGZLM1-5 cells. All the BGZLM1-5 cells were killed after heating to 100 °C for 30 min, and this treatment was used to obtain the heat-killed BGZLM1-5 postbiotic. Moreover, to obtain a heat-treated supernatant of BGZLM1-5, a part of the filtered overnight supernatant was heated at 100 °C for 30 min. The determination of the number of live bacteria used for the production of all postbiotics was performed by plating the bacteria collected before the filtration and high-temperature treatment of bacterial PBS suspension or supernatant on GM17 agar and incubating for 24 h. All the treatments (ON BGZLM1-5, SN BGZLM1-5, heat-killed BGZLM1-5, and heat-treated SN BGZLM1-5 and supernatant after 21 days of storage at 4 °C) used for the comparison of the antilisterial activity were prepared from the same number of viable bacteria.

### 2.7. Growth Kinetics and Antimicrobial Activity of Live Enterococcus faecium BGZLM1-5

The kinetics of the antimicrobial activity were followed by the inoculation of 1000 mL of GM17 broth with 1% (*v*/*v*) ON BGZLM1-5 and incubation at 37 °C [32]. Samples were taken every two hours from 0 to 24 h to determine the OD_600_, pH, and antimicrobial activity by the agar well diffusion assay (using 50 μL aliquots of samples), with *L. monocytogenes* ATCC19111 as the strain indicator.

### 2.8. Effects of Different Temperatures and pH Values on Antimicrobial Activity of Enterococcus faecium BGZLM1-5

*Enterococcus faecium* BGZLM1-5 was cultivated in GM17 broth at 37 °C for 24 h. The supernatant was obtained by centrifugation at 5000 rpm at 4 °C for 20 min and used to carry out the following studies. The antimicrobial activity of SN treated with different temperatures was investigated with minor modifications [33]. The supernatant was treated at 50, 60, 70, 80, 90, and 100 °C for 10, 30, and 60 min. The stability of SN to pH (ranging from 1 to 12) was tested with minor modifications according to [34]. After incubation at 37 °C for 2 h, the pH was adjusted to pH 7, and the antibacterial activity was examined.

### 2.9. Purification of Antimicrobial Compound

The antimicrobial compound was isolated and partially purified from 1000 mL cultures according to the method described in [6] with some modifications. Briefly, the strain BGZLM1-5 was grown in 100 mL of GM17 broth for 24 h. One liter of GM17 broth was inoculated by a 1% (*v*/*v*) ON BGZLM1-5 culture and incubated for 24 h at 37 °C. The partial purification of the antimicrobial compound of *En. faecium* BGZLM1-5 started by centrifugation at 7000 rpm for 20 min at 4 °C. The SN was precipitated using refrigerated 20%, 40%, 60%, and 80% saturated ammonium sulfate solutions, and a semipermeable membrane (cutoff, 3.5 kDa) was used for dialysis. The antibacterial activity of the crude extracts was tested and used for further purification by mixing with chloroform (1:1 *v*/*v*). Following the subsequent centrifugation (7000 rpm at 4 °C for 20 min), chloroform–aqueous interfacial proteins were recovered and resuspended in sodium phosphate buffer (pH 7). Tris-Tricine SDS PAGE (Sodium Dodecyl Sulfate polyacrylamide gel electrophoresis) 15% acrylamide gel was used to determine the molecular mass of the bacteriocin [35]. By comparing the position of the BGZLM1-5 bacteriocin zone of inhibition (gel overlaid with GM17 soft agar containing *L. monocytogenes* ATCC19111) with the nisin zone of inhibition from *Lactococcus lactis* (Sigma, Hertfordshire, UK), the molecular weight of the BGZLM1-5 bacteriocin was determined.

### 2.10. Effects of Bacteriocin-Containing Supernatant and Partially Purified Bacteriocin Produced by Enterococcus faecium BGZLM1-5 on Abundance of Listeria Monocytogenes ATCC19111 in Milk Model

A milk model was used to examine the effect of the antimicrobial compound on the abundance of *L. monocytogenes* ATCC19111 according to [36] with a minor modification. Briefly, autoclaved reconstituted skimmed milk (RSM) was separately inoculated with 10^6^ CFU/mL of *L. monocytogenes* ATCC19111. Different amounts of BGZLM1-5 bacteriocin-containing supernatant were added at final concentrations of 1%, 3%, and 5% (*v*/*v*), or the partially purified BGZLM1-5 bacteriocin was added at final concentrations of 0.1%, 0.3%, and 0.5% (*v*/*v*). The milk samples were monitored during 7 days of storage at 4 °C. Each of the two independent experiments was performed in triplicate. Principal component analysis (PCA) was performed on a number of *L. monocytogenes* ATCC19111 cells, non-treated (control) or after treatments with different concentrations of partially purified bacteriocin (0.1%, 0.3%, and 0.5%) and bacteriocin-containing supernatant (1%, 3%, and 5%), counted on different days during the storage and visualized using RStudio v2022.07.2. The R stats package and ggbiplot were used for the analysis.

### 2.11. Statistical Analysis

All the experiments were independently repeated at least twice and were conducted in triplicate. The data are presented as mean values ± standard deviations for all the independent experiments. Two-way ANOVA with the Bonferroni post hoc test was used for multiple group comparisons. Values of *p* < 0.05 were considered to be statistically significant. The statistical analysis was carried out and graphs were prepared using GraphPad Prism 10 software.

## 3. Results

### 3.1. Enterococcus faecium BGZLM1-5 Was Predicted as a Non-Human Pathogen

The *En. faecium* BGZLM1-5 WGS revealed that the genome size was 2,809,193 bp (258 contigs) comprising 2.751 protein-coding sequences (CDS), 64 tRNAs, and 1 rRNA operon, with a guanine-cytosine (GC) content of 37.9% (Figure 1). Antimicrobial resistance (AMR) analysis was performed using ResFinder (v4.1), leading to the identification of two genes associated with antibiotic resistance. These two genes correspond to a variant of *aac(6′)-Ii*, which is implicated in aminoglycoside resistance, and a homolog of *msr(C)*, involved in MLS resistance—macrolides, lincosamides, and streptogramin B. Additionally, a disinfectant resistance-related gene, *clpL*, was detected (Appendix A). Virulence factor screening was performed using the Virulence Factor of Bacterial Pathogens Database (VFDB). The analysis discovered that the BGZLM1-5 genome contains genes responsible for adhesion (*ebpA, ebpB, ebpC, srtC, ecbA, efaA,* and *sgrA*), antiphagocytosis (*cpsA/uppS* and *cpsB/cdsA*), biofilm formation (*bopD*), enzymes (*stp*), iron uptake (*vctC*), surface protein anchoring (*igT*), and immune evasion (Appendix A). According to PathogenFinder (v1.1.), BGZLM1-5 was predicted as a non-human pathogen (the probability of being a human pathogen was 0.257). The following four plasmid sequences were uncovered: rep18a (99.79%), rep29 (100%), repUS15 (99.04%), and rep2 (99.93%), as detailed in Appendix A.

### 3.2. Enterococcus faecium BGZLM1-5 Contains Gene Clusters Predicted to Encode the Production of Four Antimicrobial Peptides

The desirable traits of the probiotic strains include the production of antimicrobial peptides. An exploration of the *En. faecium* BGZLM1-5 genome was conducted to identify genes encoding bacteriocins and post-translationally modified peptides with non-bactericidal properties, utilizing the BAGEL4 online tool. The analysis through BAGEL4 revealed that the strain BGZLM1-5 possesses four candidate genes for the production of antimicrobial peptides (enterolysin A, enterocin L50, bacteriocin 31, and bacteriocin 32) (Appendix A).

### 3.3. Analysis of Bacteriocin Gene Expression of Enterococcus faecium BGZLM1-5

To show which of the bacteriocin genes is responsible for the antimicrobial activity of BGZLM1-5, gene expression analysis was performed for the following genes: *entlA*, *bac31*, *bac32*, and *entL50*. The analysis of the bacteriocin 31 transcription revealed a 16.6-fold increase at the 6th and 17.4-fold increase at the 24th h for the *bac31* gene, while the bacteriocin 32 transcription revealed a 16.4-fold increase at the 24th h for the *bac32* gene. The mRNA relative levels of enterocin L50 were significantly increased at the 6th and 24th h (5.7-fold and 8.2-fold, respectively). No changes were observed in enterolysin A (Figure 2).

### 3.4. Antimicrobial Potential of Enterococcus faecium BGZLM1-5

Further, we examined the antimicrobial potential of *En. faecium* BGZLM1-5 by diffusion tests with wells on 24 indicator strains (Table 1). Based on the results, it can be concluded that the BGZLM1-5 strain shows antimicrobial activity against eight strains. The proteinaceous nature of the antimicrobial compounds was detected against *L. monocytogenes* ATCC19111, *L. ivanovii* ATCC19119, *L. innocua* ATCC33090, and *En faecium* DDE4, while growth inhibitions of *Proteus mirabilis* ATCC12453, *Proteus mirabilis* TR4, *Aeromonas veronii* ASII-1, and *Morganella morganii* ASIII-2 were achieved by non-proteinaceous antimicrobial compounds. To examine whether the antimicrobial compound is bound to the cell or released into the supernatant, the supernatant of the *En. faecium* BGZLM1-5 overnight culture was tested (Figure 3). The supernatant and cells of the *En. faecium* BGZLM1-5 overnight culture were previously separated by filtration through a 0.22 μm filter, and the obtained cells were washed and resuspended in PBS. Additionally, the *En. faecium* BGZLM1-5 cells and supernatant were treated by heating at 100 °C for 30 min. The results showed that antimicrobial molecules are released into the supernatant and that the antimicrobial activity of the supernatant is partially resistant to high temperatures and retained over three weeks of storage at 4 °C. In contrast to the supernatant, the cell fraction lost antimicrobial activity after the high-temperature treatment.

### 3.5. Kinetics of Bacteriocin Production

To examine the kinetics of the BGZLM1-5 bacteriocin production, the antilisterial activity of live bacteria was monitored by a well diffusion assay for 24 h (Figure 4). The antimicrobial effect reaches its highest value in the 10th h. Additionally, the change in the pH value of the supernatant at 37 °C was examined during 24 h of BGZLM1-5 growth. It was shown that the pH value of the culture decreases rapidly in the exponential phase (from 6.46 ± 0.3 to 4.68 ± 0.4), while in the stationary phase, the value is maintained until the end of the test (from 4.49 ± 0.2 to 4.45 ± 0.5) (Figure 4). Additionally, it was shown that the exponential phase of BGZLM1-5 lasted from the 4th to the 8th h (OD measured at 600 nm increased from 0.29 ± 0.1 to 0.59 ± 0.08). After the eighth hour, the OD values are retained for up to 24 h.

### 3.6. Bacteriocin-Containing Supernatant of Enterococcus faecium BGZLM1-5 Retains Activity at a Variety of Temperatures and pH Levels

As the proteinaceous nature of the compounds with an antilisterial effect in the supernatant produced by the *En. faecium* BGZLM1-5 strain was demonstrated by the diffusion assay, we further tested the biochemical properties of this supernatant. The supernatant was treated at different temperatures (50, 60, 70, 80, 90, and 100 °C) for 10, 30, and 60 min and in the pH range from 1 to 12. The results showed that the bacteriocin-containing supernatant retained antilisterial activity when exposed to 50, 60, and 70 °C for 10 and 30 min and at 70 °C for 60 min. The exposure of the supernatant to 80, 90, and 100 °C for 10, 30, and 60 min decreased the antilisterial activity of the supernatant to 75% (Figure 5A). The influence of the pH value on the antilisterial activity of the *En. faecium* BGZLM1-5 bacteriocin-containing supernatant was tested by adjusting the pH of the supernatant in the range from 1 to 12, at an interval of one pH unit. The pH-sensitivity test showed that the bacteriocin antilisterial activity was not entirely lost owing to exposure to different pH levels. The antimicrobial activity maintained relative stability at pH 3–11 and retained 62.5% of the initial activity compared with that of the control after treatment at pH 1, 2, and 12 (Figure 5B).

### 3.7. Partial Purification and Determination of Molecular Weight of Enterococcus faecium BGZLM1-5 Bacteriocin

Protein precipitation by ammonium sulfate is the most usual method used to purify proteins from culture broths, so we decided to use this method for the isolation of the bacteriocin produced by *En. faecium* BGZLM1-5. Based on the results of the experiment, it can be concluded that bacteriocin is present in all the examined fractions but in different amounts. The fraction with the highest bacteriocin amount is obtained by precipitation at 40% saturation with ammonium sulfate, which can be deduced from the size of the zone of inhibition given by this fraction. Precipitates obtained at 60% and 80% ammonium sulfate saturation also contain a significant amount of bacteriocin, while in the supernatants of all the tested fractions, the concentration of bacteriocin is low, as indicated by the very narrow inhibition zone (Figure 6A). The evident molecular size of the partially purified bacteriocin produced by BGZLM1-5 was analyzed using 15% Tricine-PAGE. The gel overlaid with the indicator strain *L. monocytogenes* ATCC19111 displayed a clear zone of inhibition that was the same as the zone of inhibition of the nisin derived from *Lc. lactis*, which had a molecular weight of approximately 3.5 kDa (Figure 6B). Therefore, the molecular weight of the bacteriocin produced by BGZLM1-5 was about 3.5 kDa.

### 3.8. Application of Bacteriocin-Containing Supernatant and Partially Purified Bacteriocin of Enterococcus faecium BGZLM1-5 Effectively Reduces Abundance of Listeria monocytogenes ATCC19111 in a Milk Model

Considering these results, in the next step, we examined the effects of the bacteriocin-containing supernatant (Figure 7A) and partially purified bacteriocin (Figure 7B) on the abundance of *L. monocytogenes* ATCC19111 in the milk model over a seven-day storage period at 4 °C. After three days of storage with 3% (*v*/*v*) and 5% (*v*/*v*) bacteriocin-containing supernatants, decreases were observed in the number of *L. monocytogenes* ATCC19111 by 23.5% and 63.5%, respectively. These reductions remained unchanged until the 7th day, when reductions were observed in the number of *L. monocytogenes* ATCC19111 by 33.2% and 56.5%, respectively. In comparison with the control, only the highest concentration of partially purified bacteriocin reduced the number of *L. monocytogenes* ATCC19111 to 58.9% after three days of storage. Additionally, the same concentration was effective after the 7th day of application, and a 61% reduction was observed compared to the control. Finally, principal component analysis confirmed the strong separation of the milk samples only infected with *L. monocytogenes* ATCC19111 (control) from the infected milk samples treated with the highest concentration of partially purified bacteriocin (bacteriocin 0.5%) and bacteriocin-containing supernatant (supernatant 5%) of BGZLM1-5 (Figure 7C).

## 4. Discussion

In a modern and fast-paced lifestyle, there is a tendency to consume food without or with minimum technological processing to preserve its nutritional value [37]. Poor manufacturing practices, such as poor sanitation and contaminated processing environments, might result in food contamination by *L. monocytogenes* in the food supply chain [38]. To reduce the risk of food contamination and the spread of infection with this pathogen, the regulations of the World Health Organization (WHO) and the Food and Agriculture Organization (FAO) do not allow the presence of *L. monocytogenes* in food [39]. In recent years, great efforts have been made by the scientific community to find alternative, natural agents in food preservation [40]. The use of LAB in biopreservation is known; however, a small number of bacteriocins can be successfully applied [11,41]. Although *Enterococcus* species do not have a GRAS status, they are used as nonstarter lactic acid bacteria (NSLAB) in a variety of artisan cheeses produced in Southern Europe [42]. Their controversial nature varies from pathogens to probiotics, and it is necessary to test each strain before it is used in the food industry [43].

One of the objectives of this study was to utilize a WGS method to evaluate the safety and probiotic capabilities of *En. faecium* BGZLM1-5, a strain known for its production of bacteriocins. Enterococci exhibit genome sizes spanning from 2.5 to 3.6 Mb accompanied by a GC content that fluctuates between 37% and 45%. As observed previously, commensal strains of enterococci tend to possess smaller genomes in contrast to clinical isolates. Genetic diversity in clinical isolates arises from the incorporation of exogenous genetic material [44]. Distinctions are observed between pathogenic strains Aus0004, Aus0085, and DO and the probiotic strain T110. According to the results of a previous study [45], the genome size of the *En. faecium* strains vary substantially, and the differences relate to the presence of unique IS elements, transposons, phages, plasmids, genomic islands, and inherent and acquired antibiotic-resistance determinants. The genome of BGZLM1-5 consists of 2.81 Mb and five plasmids, housing a total of 2751 protein-coding genes. The analysis of the draft genome sequence using different bioinformatic tools indicates that *En. faecium* BGZLM1-5 is not pathogenic. However, within BGZLM1-5, the intrinsic resistance gene *msr(C)* that confers resistance to macrolides, *aac(6′)-Ii*, encoding an aminoglycoside acetyltransferase responsible for low-level aminoglycoside resistance, and a disinfectant resistance-related gene, *clpL*, are present. Earlier studies have revealed that antimicrobial resistance (AMR) genes commonly linked to pathogenic enterococci, such as vancomycin-resistance genes [46], are absent in BGZLM1-5. Furthermore, the identification of virulence genes holds significance in the characterization of enterococci strains as probiotics. Upon the investigation of the genome, genes linked to adhesion, biofilm formation, and antiphagocytosis were identified in BGZLM1-5. Past studies have revealed a correlation between the presence of these genes and the strain’s colonization ability. Adhesion is widely recognized as crucial for commensal and probiotic enterococci, facilitating colonization, persistence, and evasion of elimination by the host [43]. Using BAGEL4 software, four bacteriocins (enterolysin A, enterocin L50, bacteriocin 31, and bacteriocin 32) were found in the BGZLM1-5 genome. According to the classification by Franz et al., bacteriocin 31 and enterocin L50 belong to Class IIa enterocins of the pediocin family, while bacteriocin 32 belongs to Class IIc, other linear nonpediocin-like enterocins. Enterolysin A is a large, heat-labile bacteriocin active against LAB, *Listeria*, and *Staphylococcus* spp. and is classified in class IV bacteriocins [9]. Although the presence of multiple of bacteriocin genes is considered as a good trait for a strain to have, this does not necessarily mean that these genes are expressed simultaneously [47]. The ability to produce multiple bacteriocins is a common feature among enterococci. In this study, using qPCR, it was shown that BGZLM1-5 expresses three bacteriocins (bacteriocin 31, bacteriocin 32, and enterocin L50) at the mRNA level.

*Enterococcus* species exhibit diverse bacteriocins known as enterocins, which possess specific antimicrobial properties aimed at inhibiting the growth of species closely related to the producing bacteria [48]. Their extensive study is attributed to their effectiveness against certain foodborne pathogens and spoilage bacteria [10]. The majority of enterocins belong to class II bacteriocins and are characterized as unmodified and resistant to high temperatures [9]. Our study has revealed that BGZLM1-5 exhibits a wide-ranging efficacy against Gram-positive and Gram-negative bacteria. The antimicrobial activity against enterococci and *Listeria* sp. is attributed to proteinaceous compounds, whereas non-proteinaceous compounds are responsible for the antimicrobial activity against Gram-negative bacteria. Additionally, enterococci are recognized for their antimicrobial impact through the production of lactic acid, hydrogen peroxide, carbon dioxide, and diacetyl in addition to bacteriocins [49]. The results obtained in this study, pointing to the stability of the antimicrobial properties of the bacteriocin-containing supernatant of BGZLM1-5, are characteristic of enterocins produced by various *En. faecium* strains that have already been characterized [50]. We further examined the kinetics of bacteriocin production, and it was shown that the kinetic of bacteriocin production by BGZLM1-5 is in accordance with previously published data showing the more rapid production of enterocins in the initial stages of cell growth, and the decrease in enterocins production with increasing incubation time and cell density in the medium [51]. These authors assumed that such kinetics could be a part of several regulated mechanisms, such as a quorum-sensing mechanism or an induction factor, a histidine kinase. Furthermore, another study [52] assumed that low pH, as well as the presence of proteolytic enzymes in the medium, could decrease antimicrobial activity owing to the degradation of the bacteriocin after a longer incubation time. The molecular weight of the bacteriocin produced by BGZLM1-5 was about 3.5 kDa, which corresponds to previous results obtained for other bacteriocins produced by *En. faecium* [9,53]. Chakchouk-Mtibaa et al. [54] purified the active peptide from the cell-free supernatant of *En. faecium* FL31, and the results revealed a single band with an estimated molecular mass of approximately 3.5 kDa, which is similar to the molecular mass of the partially purified bacteriocin produced by *En. faecium* BGZLM1-5 in our study. After the partial purification of the *En. faecium* 130 bacteriocin by the adsorption–desorption technique, and the analysis by Tris-Tricine SDS-PAGE, the results showed a molecular mass from 3.5 to 6.5 kDa and indicated that the bacteriocin probably belongs to class IIa enterocins. Additionally, the partially purified bacteriocin of BGZLM1-5 showed antimicrobial activity against *L. monocytogenes* ATCC19111, *L. ivanovii* ATCC19119, *L. innocua* ATCC33090, and related species, such as *En faecium* DDE4, which is a characteristic feature of class IIa bacteriocins. Although the precipitate obtained after saturation with 40% ammonium sulfate was used for further purification because it had the highest amount of bacteriocin, we noticed that bacteriocin(s) was/were also present in other precipitates obtained after saturation with 60% and 80% ammonium sulfate. In a previous study, when bacteriocin was purified, more than one band was shown to have activity [55]. The utilization of bacteriocins in milk and subsequent heat treatment is a feasible approach for eradicating *L. monocytogenes*. In this study, we demonstrated that the application of the bacteriocin-containing supernatant and partially purified bacteriocin at the highest concentrations (5% *v/v* and 0.5% *v*/*v*, respectively) exhibited the capability to decrease the number of *L. monocytogenes* ATCC19111 in the milk model after three days of storage. By applying the highest concentration of the bacteriocin-containing supernatant, the reduction in the number of *L. monocytogenes* ATCC19111 was maintained during all seven days of storage at the refrigeration temperature, while this was not the case with the partially purified bacteriocin. A possible reason for this is that in the supernatant, in addition to the presence of bacteriocins, there are also non-proteinaceous compounds, such as lactic acid or hydrogen peroxide, that can have antimicrobial activity. Reductions in the numbers of viable of *L. monocytogenes* in milk and dairy products have been shown in other studies [56,57]; however, the use of strains that produce enterocins in situ has a more significant effect on the reduction in the number of pathogenic bacteria [58]. According to these results, bacteriocins produced by BGZLM1-5 could be considered as promising antilisterial agents. Considering all the other characteristics of the BGZLM1-5 products demonstrated in our article, such as extreme temperature and pH resistance, BGZLM1-5 could be safely used for antilisterial prevention in dairy products. Additionally, the possibility of the usage of live BGZLM1-5 in dairy food production should not be rejected, as this strain is characterized as non-pathogenic and without the potential for antibiotic-resistance spreading. The approach for using enterococcal strains, proven to be safe, in food production could not only bring the benefit of continuous bacteriocin production but also improve products’ taste and technological characteristics.

## 5. Conclusions

*Listeria monocytogenes*, a human pathogen, is widespread and can disseminate and contaminate food products. Bacteriocins produced by enterococci have been especially studied because of their strong activity against most Gram-positive pathogens. In this study, the autochthonous *En. faecium* BGZLM1-5, with strong antilisterial activity, was characterized as a non-human pathogen after genome sequencing and bioinformatics analyses. The bacteriocin-containing supernatant was thermally stable, and the antimicrobial activity retained 75% of the initial activity compared with that of the control after treatment at 90 °C for 30 min. The antimicrobial activity maintained relative stability at pH 3–11 and retained 62.5% of the initial activity compared with that of the control after treatment at pH 1, 2, and 12. The highest activity of the partially purified bacteriocin was obtained after precipitation at 40% saturation with ammonium sulfate and further purification by mixing with chloroform. The application of 3% and 5% (*v*/*v*) bacteriocin-containing supernatants decreased the number of viable *L. monocytogenes* ATCC19111 by 23.5% and 63.5%, respectively, in milk after three days of storage. Compared with the control, the 0.5% partially purified bacteriocin significantly decreased (by 58.9%) the number of viable *L. monocytogenes* ATCC19111 in milk after three days of storage. Future studies should examine the efficacy of both the live strain BGZLM1-5 and the isolated bacteriocin in reducing the abundance of *L. monocytogenes* in other food matrices to determine the wider applicability of this strain in different food products.

## Figures and Tables

**Figure 1 microorganisms-11-02844-f001:**
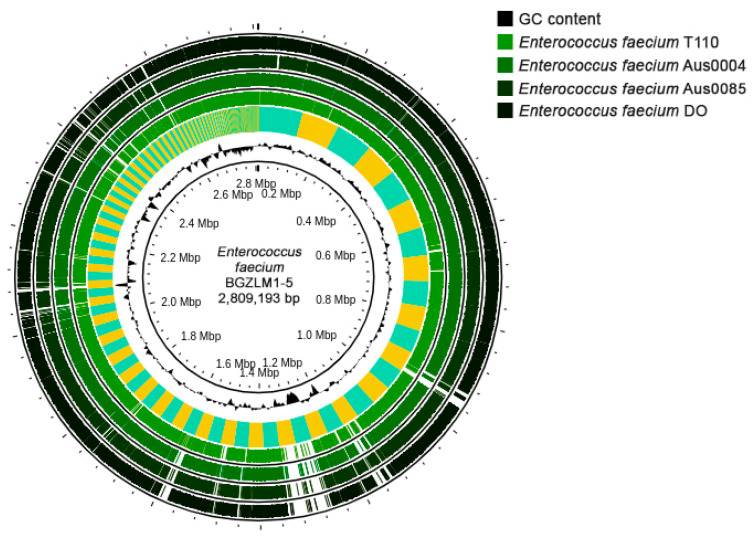
Circular genome map of *Enterococcus faecium* BGZLM1-5 visualized and compared with those of selected pathogenic and probiotic *Enterococcus faecium* strains using Proksee. Circles from the inside to the outside: the first ring represents the scale marks in megabase pairs (Mbp) and GC content (black); the second ring represents a comparative analysis of the BGZLM1-5 genome against the strain T110 genome using BLAST+ 2.12.0 software, followed by rings representing the strain Aus0004, Aus0085, and DO compare genomes.

**Figure 2 microorganisms-11-02844-f002:**
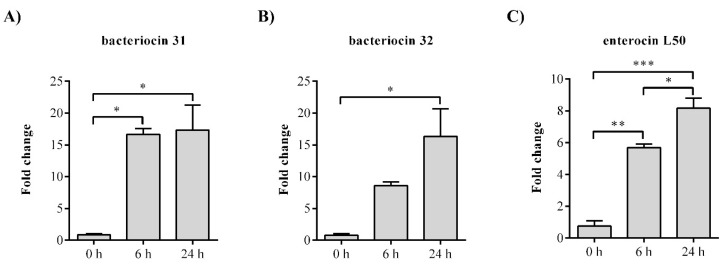
The relative expression levels of bacteriocin genes from *Enterococcus faecium* BGZLM1-5. Statistical significance was tested using one-way ANOVA followed by the Dunnett post hoc test in GraphPad 10. Statistical significances *p* < 0.0332, *p* < 0.002, and *p* < 0.0002 are marked as ***, **, and * respectively.

**Figure 3 microorganisms-11-02844-f003:**
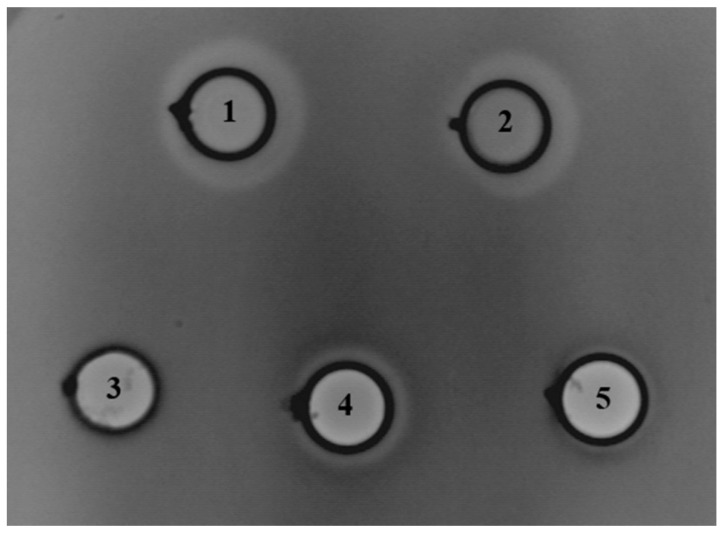
Direct inhibitory effect of *Enterococcus faecium* BGZLM1-5 toward *Listeria monocytogenes* ATCC19111: overnight culture (1), supernatant (2), heat-treated cells (3), heat-treated supernatant (4), supernatant after three weeks’ storage at 4 °C (5). The dots indicate the places where the crystals of pronase E were placed to test the proteinaceous nature of the antimicrobial compounds produced by *En. faecium* BGZLM1-5.

**Figure 4 microorganisms-11-02844-f004:**
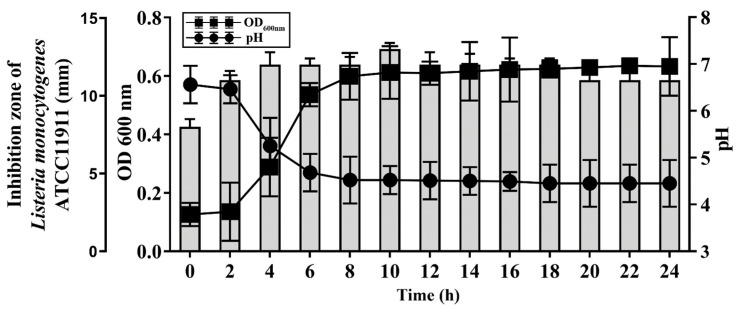
Antilisterial effect of *Enterococcus faecium* BGZLM1-5 during growth in GM17 broth and bacteriocin activity. Growth of *En. faecium* BGZLM1-5 was determined spectrophotometrically (OD 600 nm). Circles in the graph indicate pH, while squares indicate the optical density (OD) of the *En. faecium* BGZLM1-5 culture. Bars indicate the activity (mm) of enterocin against the indicator strain *L. monocytogenes* ATCC19111.

**Figure 5 microorganisms-11-02844-f005:**
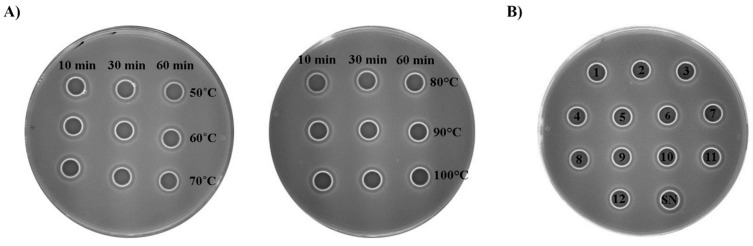
Effects of different temperatures (**A**) and pH values (**B**) on *Enterococcus faecium* BGZLM1-5 bacteriocin activity. Antimicrobial activity of overnight supernatant (SN) BGZLM1-5 culture after treatment at different temperatures for 10, 30, and 60 min and after treatment at different pH levels in the range from 1 to 12 (numbers inside the circles correspond to pH values) for the indicator strain *L. monocytogenes* ATCC19111.

**Figure 6 microorganisms-11-02844-f006:**
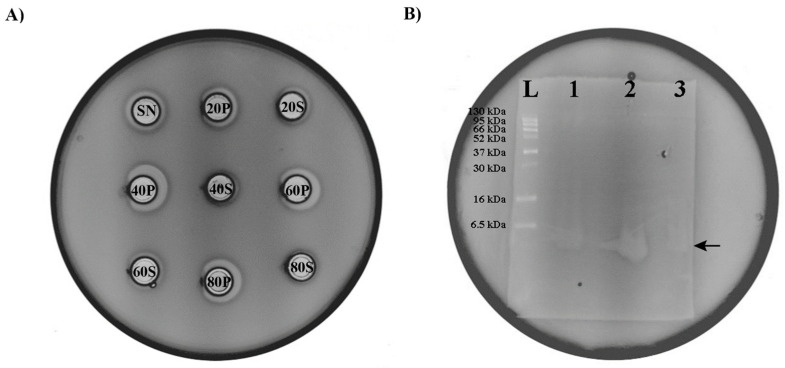
Purification and determination of bacteriocin size. (**A**) Antimicrobial activity of *En. faecium* BGZLM1-5 supernatant fractions obtained by ammonium sulfate precipitation against the indicator strain *L. monocytogenes* ATCC19111. SN—supernatant of overnight culture of *En. faecium* BGZLM1-5 purified from cells by filtration through a 0.22 μm filter; S—Supernatant; P—Precipitate; 20, 40, 60, and 80 are percentages of saturation with ammonium sulfate. (**B**) Determination of bacteriocin size; L—protein ladder (6.5 kDa–270 kDa); 1—nisin derived from *Lc. lactis*; 2—precipitate saturated with 40% ammonium sulfate; 3—precipitate obtained after chloroform treatment.

**Figure 7 microorganisms-11-02844-f007:**
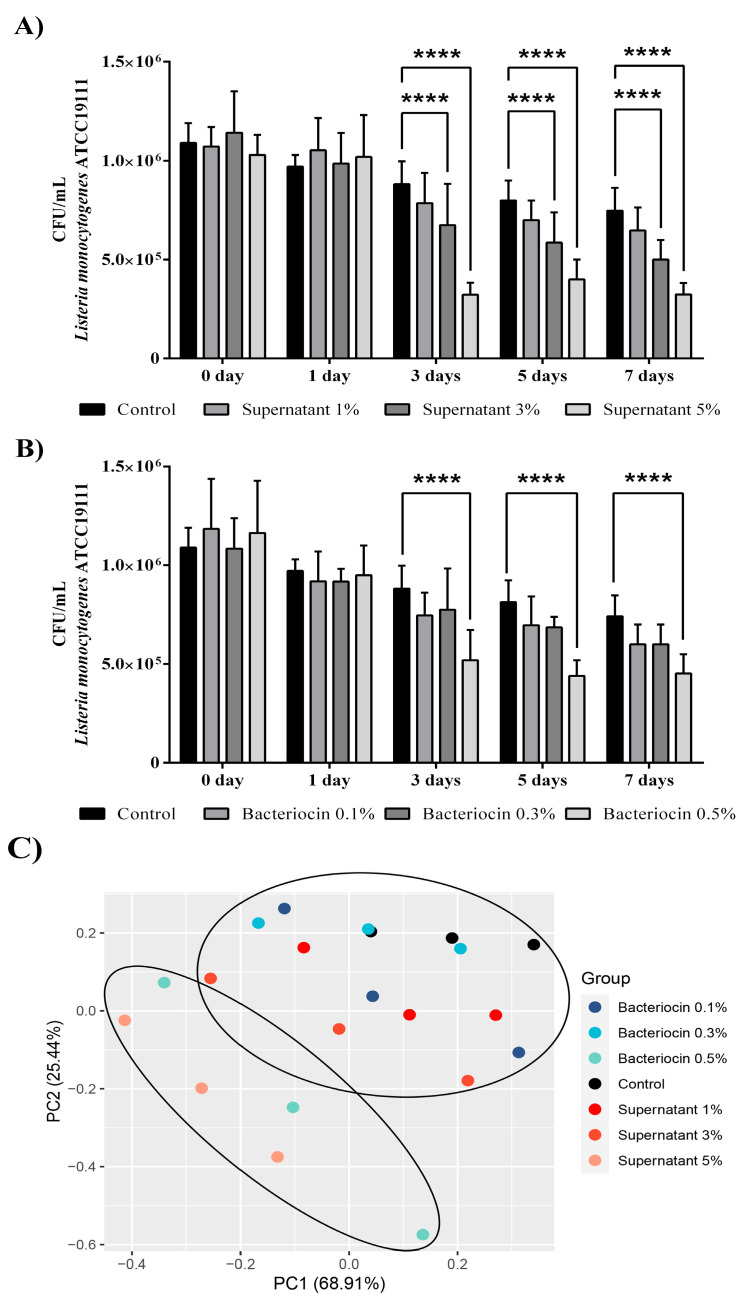
Reduction in the number of *Listeria monocytogenes* ATCC19111 in milk treated with bacteriocin-containing supernatant (**A**) and partially purified bacteriocin (**B**) produced by *Enterococcus faecium* BGZLM1-5. Principal component analysis (PCA) of *Listeria monocytogenes* ATCC19111 abundance in milk samples non-treated (control) or treated with different concentrations of partially purified bacteriocin (0.1%, 0.3%, and 0.5%) and bacteriocin-containing supernatant (1%, 3%, and 5%) on different days during the storage **(C)**. Statistical significance was tested using two-way ANOVA followed by Bonferroni post hoc test in GraphPad 10; *p* < 0.0001 is marked as ****.

**Table 1 microorganisms-11-02844-t001:** Indicator strains used in this study and inhibitory activities of overnight cultures of live *Enterococcus faecium* BGZLM1-5.

Indicator Strain	Reference	Activity
*Bacillus cereus* ATCC11778	ATCC **	-
*Bacillus subtilis* subsp. *spizizenii* ATCC6633	ATCC	-
*Escherichia coli* ATCC25922	ATCC	-
*Listeria monocytogenes* ATCC19111	ATCC	+ *
*Listeria ivanovii* ATCC19119	ATCC	+ *
*Listeria innocua* ATCC33090	ATCC	+ *
*Proteus mirabilis* ATCC12453	ATCC	+
*Pseudomonas aeruginosa* MMA83	[15]	-
*Salmonella enterica* subsp. *enterica* serovar Typhimurium ATCC14028	ATCC	-
*Staphylococcus aureus* ATCC25923	ATCC	-
*Staphylococcus epidermidis* ATCC12228	ATCC	-
*Yersinia enterocolitica* ATCC27729	ATCC	-
*Escherichia coli H7:0157* ATCC35150	ATCC	-
*Proteus mirabilis* TR4	[16]	+
*Enterococcus faecium* DDE4	[17]	+ *
*Streptococcus pyogenes* A2941	Labodijagnostika, Belgrade, Serbia	-
*Klebsiella pneumoniae* Ni9	[18]	-
*Aeromonas veronii* ASII-1	[16]	+
*Acinetobacter baumannii* 6077/12	[16]	-
*Morganella morganii* ASIII-2	[16]	+
*Salmonella enterica* serovar Enteritidis E657/7	Veterinary Institute, Novi Sad, Serbia	-
*Providencia alcalifaciens* AAI-1	[16]	-

Notes: + presence of the inhibition zone, - no inhibition, *—antimicrobial compound is proteinaceous. ** ATCC—American Type Culture Collection.

**Table 2 microorganisms-11-02844-t002:** List of the primers used in this study.

Primer Name	Sequence 5′-3′	Reference
EntlA_Fw	GGACAACAATTCGGGAACACT	[29]
EntlA_Rw	GCCAAGTAAAGGTAGAATAAA	
EntL50_Fw	TGGGAGCAATCGCAAAATTAG	[30]
EntL50_Rw	ATTGCCCATCCTTCTCCAAT	
Bac31_Fw	AGCAACTTATTATGGAAATGGTG	This study
Bac31_Rev	AACGGATCCTTTCTATCTAGGAGCCC	This study
Bac32_Fw	ATTCACCCCTTCTGTTTCATTTTCTC	This study
Bac32_Rev	TTTAAGCTTACTAAATGTAGTAATAATATTTGGC	This study
RecA_Fw	TTCTTTAGCGTTAGATGTTG	[31]
RecA_Rw	CCTTCTTGGGAAATACCTT	

## Data Availability

The data supporting this study’s findings are available from the corresponding author (N.P.) upon reasonable request. The preliminary genome sequences of BGZLM1-5 have been submitted to GenBank under the accession JAVDBU000000000. The version described in this paper is version JAVDBU010000000.

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
