# Peer review of "Insight into the Postbiotic Potential of the Autochthonous Bacteriocin-Producing Enterococcus faecium BGZLM1-5 in the Reduction in the Abundance of Listeria monocytogenes ATCC19111 in a Milk Model"

_microorganisms, 2023, doi:10.3390/microorganisms11122844_

Round 1
Reviewer 1 Report
Comments and Suggestions for Authors
This article “Insight into the postbiotic potential of the autochthonous bacteriocin-producing Enterococcus faecium BGZLM1-5 in the reduction of Listeria monocytogenes ATCC19111 in a milk model” presents a detailed study on the Enterococcus faecium BGZLM1-5 strain, which was isolated from raw cow milk obtained from households in the Zlatar Mountain region. The research shows that this strain exhibits robust antilisterial activity that remains unaffected even after exposure to high temperatures, refrigeration, and across a wide pH range. The study also includes genomic and in silico analyses that reveal the non-pathogenicity of the BGZLM1-5 strain to humans but notes the presence of genes conferring resistance to macrolides and aminoglycosides. The article highlights the postbiotic potential of this strain in reducing Listeria monocytogenes in a milk model and its potential application in food safety. The strong point of this article is the presentation of a detailed research on the Enterococcus faecium BGZLM1-5 strain, including its probiogenomic characteristics, antilisterial activity, and potential application in food safety. The study also includes genomic and in silico analyses that reveal the non-pathogenicity of the BGZLM1-5 strain to humans, which is valuable information for risk assessment. Additionally, the article highlights the postbiotic potential of this strain in reducing Listeria monocytogenes in a milk model, which may have important implications for the dairy industry and food safety in general.
The study focuses on a single bacterial strain and a milk model, which may limit the applicability of the results to other food contexts. One possible experiment that could complement this study would be to investigate the efficacy of the BGZLM1-5 strain in reducing Listeria monocytogenes in other food matrices, such as meat or vegetables. This would help to determine the broader applicability of this strain in different food products and provide more comprehensive data on its potential as a postbiotic agent. Additionally, further studies could explore the safety and stability of the bacteriocin-containing supernatant and partially purified bacteriocin under different storage conditions and processing methods. These experiments would help to address some of the limitations of the current study and provide more robust data on the potential of this strain in food safety applications.
The introduction is too long, as well as the paragraphs are huge, this needs to be better. Please improve the abstract to cover the important topics reviewed and discussed in this article. The abstract is written in a way lacks logic. It should highlight the salient findings more critically;
The results have long paragraphs. I suggest reducing the size of the paragraphs. The results of this study are not fully explained therefore the interpretation of the results is very difficult. The author needs to provide the % increase or decrease rather than just writing ''significantly increased….'';
Authors should discuss the results integrally. The discussion is based on individual results. I suggest that integrating the results will give more value to the work. I suggest that you discuss by integrating all your results. You can use correlation tests (PCA or Pearson Correlation).
The conclusion is totally confusing. Re-write the conclusion! It needs to be much improved.
The discussion is poorly written hence, needs rewriting. The discussion should be further strengthened by adding some more relevant papers. The literature search is INSUFFICIENT, only few related research papers in the past five years are cited (39%, approximately), add the latest research results appropriately.
Reviewer 2 Report
Comments and Suggestions for Authors
Dear Editor,
The manuscript “Insight into the postbiotic potential of the autochthonous bacteriocin-producing Enterococcus faecium BGZLM1-5 in the reduction of Listeria monocytogenes ATCC19111 in a milk model” looked for a bacteriocin-producing strain of Enterococcus faecium BGZLM1-5, and the postbiotic potential of the stain of anti-listerial activity in the milk model. It would have been preferable if the work had finished the further purification. However, the manuscript could be considered for publication after minor revision.
1. I would advise the authors to utilize the most recent Microorganisms template.
2. References should be numbered in order of appearance in the text.
3. According to the abstract, the team sequenced the whole genome of BGLM1-5, and in silico research demonstrated that BGLM1-5 is not harmful to humans. In the abstract, I would suggest removing the statement "However, in silico analysis is essential before considering the application of such strains in food production, given their controversial nature."
4. When the text contains Listeria monocytogenes or Enterococcus faecium in the main body yet L. monocytogenes and Ent. faecium were located elsewhere, please utilize a consistent format.
5. The statement "Most enterococcal bacteriocins of group II are thermostable and resistant to various technological challenges, making them suitable for use in the food industry" appeared in the introductory section. Please explain what "resistant to various technological challenges" means.
6. In the MM section, under "2.2 whole genome sequencing...." The first phrase, "The complete DNA," might be improved by changing it to "The genomic DNA."
7. In MM section, "2.9 purification of antimicrobial compound". substituted "saturated" in the form of "the SN was precipitated using refrigerated 20%, 40%, 60%, and 80% saturated ammonium sulfate solution" .
8. In the results part, “3.3 analysis of bacteriocin gene expression...” remove the “sixth” in the fifth line of the paragraph.
9. In the discussion part, the authors may consider the statement “After partial purification of En. faecium 130 bacteriocin by the adsorption-desorption technique, and the analysis by sodium dodecyl sulfate-polyacrylamide gel electrophoresis (SDS-PAGE) the results showed a molecular mass of 3.5 to 6.5 kDa and indicated that bacteriocin probably belongs to the class IIa enterocins.” The bacteriocin-like substance generated by BGLM1-5 appears to belong to Class lla bacteriocin. However, it is categorized not only by the molecular weight stressed in the paper, but also by its antibacterial spectrum. According to the findings, BGLM1-5 exhibited substantial inhibitory action against Listeria spp. as well as cloase related bacteria of Enterococcus spp. This is yet another method for classifying Class IIa bacteriocins.
10. I would like to suggest that the authors replace "Class II.1." and "Class II.3." in the discussion section to "Class IIa" and "Class IIc," respectively.
11. Remove "capability" or "capacity" from the fifth paragraph of the conclusion.
Comments on the Quality of English LanguageEnglish sounds good and understandable.
Reviewer 3 Report
Comments and Suggestions for Authors
The manuscript entitled "Insight into the postbiotic potential of the autochthonous bacteriocin-producing Enterococcus faecium BGZLM1-5 in the reduction of Listeria monocytogenes ATCC19111 in a milk model" by Popović et al. provides interesting data on bacteriocin-producing probiotics and postbiotics based on them in the milk model.
I would like to recommend the paper for publication after some minor revisions and additional bioinformatical/statistical analysis.
1. The manuscript is not formatted according to the journal's requirements. There are no line numbers provided.
The introduction consists of one paragraph, the discussion consists of two paragraphs.
Please divide your manuscript into paragraphs consistently.
2. Please, rewrite the abstract and add some background at the beginning.
3. Page 4, WGS. Please, provide the library preparation protocol for WGS.
4. Page 4, WGS. Kraken software is usually applied for metagenomics data. Of course, taxonomical identification provided with Kraken could be used for bacterial species identification and search for the closest genome, but the software is not intended for this. There are a lot of bioinformatical tools that could be used to search for the reference genome based on WGS data. Please, consider applying other tools to support the Kraken outcomes and adding the results to the manuscript.
5. Page 6, Statistical analysis. ANOVA following Tukey's posthoc test usually does not have enough statistical power for the analysis of the data obtained from three biological replicates (experiments) per 'group'.
The analysis of variance assumes that the data fit the normal distribution. Please, provide the results of data normality and homogeneity of variances testing, and then perform sufficient parametric or nonparametric tests for the following statistical analysis and update the manuscript if the results change.
6. Figure 6. Please, check if the blot figure meets the journal requirements.
7. Figure 7. Please, provide an explanation of hypothesis testing results in the figure legend.
Comments on the Quality of English Language1. Please, consider replacing the "in silico analysis" term with "bioinformatical analysis" or "genomic analysis". In silico usually means compute modeling/simulation.
2. Please, consider writing "Ivanov et al." instead of "Ivanov and colleagues"/"Ivanov and co-authors".
Round 2
Reviewer 1 Report
Comments and Suggestions for Authors
Thanks for attending all the suggestions. I congratulate and inform the authors you that your submission "Insight into the postbiotic potential of the autochthonous bacteriocin-producing Enterococcus faecium BGZLM1-5 in the reduction of Listeria monocytogenes ATCC19111 in a milk model" met all expectations for publication in Microorganisms (MDPI).